# Automated Assessment of Initial Answers to Questions in Conversational Intelligent Tutoring Systems: Are Contextual Embedding Models Really Better?

Colin M. Carmon [1,2,*], Brent Morgan [3] , Xiangen Hu [1,2] and Arthur C. Graesser [1,2]

1   Institute for Intelligent Systems, University of Memphis, Memphis, TN 38152, USA;
    xhu@memphis.edu (X.H.); graesser@memphis.edu (A.C.G.)
2   Department of Psychology, University of Memphis, Memphis, TN 38152, USA
3   Department of Psychology, Rhodes College, Memphis, TN 38112, USA; brent.morgan.phd@gmail.com
*   Correspondence: cmcarmon@memphis.edu

**Abstract:** This paper assesses the ability of semantic text models to assess student responses to electronics questions compared with that of expert human judges. Recent interest in text similarity has led to a proliferation of models that can potentially be used for assessing student responses. However, it is unclear whether these models perform as well as early models of distributional semantics. We assessed 5166 response pairings of 219 participants across 118 electronics questions and scored each with 13 different computational text models, including models that use Regular Expressions, distributional semantics, embeddings, contextual embeddings, and combinations of these features. Regular Expressions performed the best out of the stand-alone models. We show other semantic text models performing comparably to the Latent Semantic Analysis model that was originally used for the current task, and in a small number of cases outperforming the model. Models trained on a domain-specific electronics corpus for the task performed better than models trained on general language or Newtonian physics. Furthermore, semantic text models combined with RegEx outperformed stand-alone models in agreement with human judges. Tuning the performance of these recent models in Automatic Short Answer Grading tasks for conversational intelligent tutoring systems requires empirical analysis, especially in domain-specific areas such as electronics. Therefore, the question arises as to how well recent contextual embedding models compare with earlier distributional semantic language models on this task of answering questions about electronics. These results shed light on the selection of appropriate computational techniques for text modeling to improve the accuracy, recall, weighted agreement, and ultimately the effectiveness of automatic scoring in conversational ITSs.

**Keywords:** computational linguistics; conversational systems; electronics training; intelligent tutoring systems; natural language processing; naval training; sentence-level semantics

## 1. Introduction

Training in electrical engineering is of paramount importance in today's technologically driven world. Electrical engineers design, develop, and maintain a wide range of electrical systems. Proper training equips engineers with the knowledge and skills necessary to tackle complex challenges in domains like electronics. It enables them to understand fundamental principles, reason, and apply problem-solving techniques. With rapid advancements in technology, effective training in electrical engineering is crucial to drive progress and meet society's evolving needs. This study investigates a natural-language-based technology that provides electrical engineering training for naval operations.

Sailors enrolled at the Navy Nuclear Power Training Center undergo comprehensive training in electronics fundamentals to successfully complete the Nuclear Field A School (NFAS) and attain the Electronics Technician Nuclear (ETN) rating. These individuals

have exceptional aptitude, as determined by their performance on the Armed Services Vocational Aptitude Battery (ASVAB; [1,2]) and the Basic Electricity and Electronics (BEE) test. Training equips them with the necessary knowledge and skills to excel in managing and maintaining complex electronic systems crucial to the Navy's nuclear power operations. The rigorous selection process ensures that only highly skilled individuals become ETN personnel. Interestingly, the primary assessment of passing training courses consists of answering electronic circuit questions in natural language that requires reasoning and that covers all expected answers to a question accurately. Consequently, accurate and complete answers in natural language is the bar that sailors are expected to pass. This is very different from an assessment that involves multiple-choice questions or actions in a simulation. Consequently, automated analyses of the answers to these questions with computational linguistics and AI is expected to have tremendous value for providing feedback on performance.

Automated conversational tutoring systems can also play a valuable role in training electrical engineers by providing interactive and personalized instruction for course curriculum materials [3,4]. These systems offer solutions to common problems in trade training environments such as accessibility, scalability, continuous learning, cost-effectiveness, and access to resources. Once again, advances in computational linguistics and natural language AI are important in developing conversational tutors that are adaptive and personalized. These NLP assessment and conversational tutoring systems rely on advances in automatic semantic processing models to assess student input. One class of semantic models consists of distributional semantics in large language models. Distributional semantic models develop methods for quantifying and categorizing semantic similarities between linguistic text segments (i.e., 1 to N words) based on their distributional properties in large samples of language data. Text segments that have similar distributions with other text segments have similar meanings. There are many contexts where distributional semantics techniques can be applied for assessing text similarity, such as automatic grading of students' text productions.

Automatic grading of natural language can be broken down according to question types: fill-in-the-blank, short answer, and essay. Fill-in-the-blank answers generally consider one key term or phrase. Automatic Short Answer Grading (*ASAG*; [5]) refers to the assessment of short natural language responses to objective questions. Unlike Automatic Essay Grading (*AEG*), which considers a balanced integration of both style and semantic content [6], ASAG typically focuses on grading shorter segments of text (one sentence to one paragraph) with a primary focus on the semantic content alone. For this reason, we can consider many modern conversational intelligent tutoring systems (*ITSs*) to be interactive forms of ASAG. Unlike ITSs, it is important to acknowledge, from a practical point of view, that humans are prone to making errors in essay grading, ASAG, and ITSs that result from fatigue and bias [7]. ITSs are potentially less costly than human tutors in terms of time invested as well [8], and, depending on the knowledge domain or task, may combat a shortage of available human tutors.

## 1.1. Intelligent Tutoring Systems

ITSs provide immediate, individualized instruction and feedback to students, sometimes without intervention from a human instructor, and implement various pedagogical strategies and methods of assessment [9]. In addition to immediate, automatic feedback and individualized instruction, ITSs minimize common assessment and consistency problems that humans frequently exhibit. ITSs that incorporate natural language communication with students are called conversational ITSs. Examples of early notable conversational ITSs on STEM topics include *Why2Atlas* [10] and *AutoTutor* [11]. Modern conversational ITSs provide instruction and feedback (hints, prompts, tutoring questions, etc.) to the student with similarities to human tutoring [12–15]. In this paper, we focus on the assessment of the students' initial responses to a main question that is part of a conversational ITS that teaches an electronics curriculum through conversation.

The ITS in the study, *ElectronixTutor* [4], teaches electronics by holding a conversation in natural language with students. The system asks students difficult questions that require approximately 1–5 sentences (called expectations) in a good answer. The tutor holds a multi-turn dialogue that guides them to articulate expected content that is missing in their initial response. The system's dialogue and assessment of student input is currently driven by a combination of Regular Expressions (*RegExes*; [16]) and Latent Semantic Analysis (*LSA*; [17]). These components are described later.

The *initial answer* represents the student's first attempt to fully answer a question. Comparing initial answers of students to good answers to electronics questions is essential for ElectronixTutor to select subsequent dialogue moves to adaptively respond to the individual student. The system's ability to branch to appropriate dialogues creates the functional scaffolding of human-like conversation in the tutorial dialogues. It is important to accurately assess initial answers to a main question because errors in assessing initial answers result in awkward dialogue branching for the remainder of the dialogue.

We can observe questions with as little as one expectation, or as many as five expectations. The following is an example of a question in ElectronixTutor, the ideal good answer, and a breakdown of the ideal answer into three expectations:

*Main Question:* How does the way the Zener diode is connected help protect a bulb in a circuit? (Figure 1)
*Ideal Good Answer:* The Zener diode is connected as a reverse-biased diode. The breakdown voltage is less than the burn-out voltage of the bulb. Extra current flows through the diode instead of the bulb.
*Expectation 1:* The Zener diode is connected as a reverse-biased diode.
*Expectation 2:* The breakdown voltage is less than the burn-out voltage of the bulb.
*Expectation 3:* Extra current flows through the diode instead of the bulb.

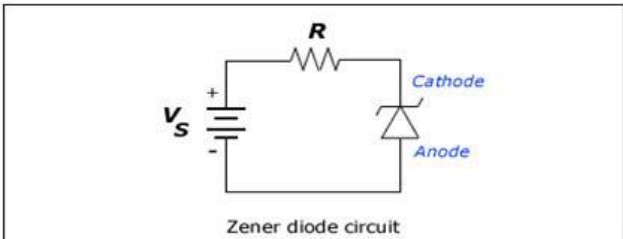

**Figure 1.** Associated figure for main question: "How does the way the Zener diode is connected help protect the bulb in a circuit?".

The conversational component of ElectronixTutor implements AutoTutor [11–13], a system that was designed to simulate human tutoring. After the student produces the initial response in natural language, the system generates several turns that give feedback on the student contributions (positive, negative, neutral) and dialogue moves to help the student fill in missing expectations or words in expectations. The dialogue moves include pumps (e.g., tell me more), hints, assertions, corrections of errors, and a variety of other dialogue moves. The feedback and dialogue moves depend on what the student has articulated. It is important to emphasize that the initial response of the student has an extremely large impact on how the multiturn dialogue evolves and branches out in different directions. Therefore, it is important to optimize the accuracy of semantic matches between the content of student answers and the expectations. This analysis of the initial responses to questions is the focus of the present study. It is beyond the scope of this article to address the conversation that evolves after the students' initial responses to questions.

### 1.2. Models for Assessing the Semantics of Students' Natural Language Answers

Systems like ElectronixTutor and AutoTutor have traditionally relied on Latent Semantic Analysis (LSA) and Regular Expressions (RegExes) for semantic models that compare students' verbal answers to expectations. However, there are other distributional seman-

tics and word embedding techniques that also compute measures for similarity between student NL answers and expectations, as will be elaborated in this section. The performances of more recent models (e.g., Word2Vec variants, BERT variants, GPT-3 variants) were compared to the performance of traditionally used techniques such as LSA and RegEx for tasks that automatically grade open-ended answers in ASAG, which are the initial answers to questions in ITS dialogues. By comparing the performances of these models, we can identify the most appropriate models for the current task of automatically grading open-ended answers in a domain-specific conversational ITS.

LSA and other similar distributional semantics techniques have been around since the 1980s [18]. More recently, several computational semantics techniques have been developed: word embedding models such as Word2Vec (*W2V*; [19,20]), transformer models such as Bidirectional Encoder Representations from Transformers (*BERT*; [21]), Sentence Bidirectional Encoder Representations from Transformers (*SBERT*; [22]), Generative Pre-trained Transformer 3 (*GPT-3*; [23]), and Sentence-Generative Pre-trained Transformer (*SGPT*; [24]) models.

Latent Semantic Analysis [17] is a distributional semantic model for assessing the similarity of pairs of texts expressed in natural language. Distributional semantic models use linear algebra to compute these representations. In the simple case of single words, "chair", "table", and "eat", for example, they often appear in the same documents and, as such, have high semantic similarity to each other. When considering text segments with multiple words, the multidimensional vectors of single words are combined into a text segment vector and compared with the combined vector of another multiword segment. Word ordering is not accounted for in LSA. Instead, the meaning of a word depends on the company of other words that surround it in documents. In the context of the current study, student contributions to electronics questions should share semantic overlap with language used in the domain-specific electronics texts that make up the training corpus for the model.

Domain-specific LSA spaces, such as electronics and electronic circuits, start by counting the number of particular words that occur in particular documents, resulting in a word-by-document matrix. The sequential order of words (alternatively called terms) appearing in documents does not matter in this bag-of-words approach. A reasonable corpus of texts is 10 million words distributed in 40,000 documents with about 100,000 word tokens; a 100,000 by 40,000 matrix has most cells being 0. A statistical method called Singular Value Decomposition (SVD) reduces the large space to 100–500 statistical dimensions. These are the dimensions that are used to assess the similarity between the student contributions and expectations. There is a vector of dimension loadings on a student contribution and another vector of dimension loadings on the expectation; the similarity of the loadings determines semantic overlap.

There are several metrics on individual words that can be derived from the word-by-document matrix and the SVD. For example, a term frequency–inverse document frequency (TFIDF) evaluates the relevance of a word to a document within a corpus. Word (term) frequency can be computed as:

$$tf_{t,d} = \frac{n_{t,d}}{\textit{\# terms in document}}$$

where *n* is the number of times a term (*t*) occurs in the document (*d*). Inverse document frequency is computed as:

$$idf_t = log \frac{\textit{\# documents in corpus}}{\textit{\# documents containing t}}$$

We can compute the TF-IDF for all words contained in the corpus where words with higher scores are more important in a specific corpus of documents than words with lower scores. Term frequency–inverse document frequency is a statistic that is intended to reflect how important a target word is to a document in a collection or corpus. TF-IDF is computed as:

$$(tf\_idf)_{t,d} = tf_{t,d} * idf_t$$

Semantic fields mirror the semantic structure extracted from an original corpus of text documents. Semantic fields exist as modified vector space representations obtained by computing word embeddings in documents in a large text corpus. Once the semantic field representation is obtained, vectors can be accessed quickly in order to compute similarities between text samples of interest, such as an ideal answer and a student response. The metric of similarity is a cosine match score from $-1$ to 1, with 0 representing no semantic similarity and 1 representing a perfect similarity between the student response and the expectation by virtue of the constructed LSA space. Given two vectors ($x$ and $y$), we can compute the cosine similarity:

$$similarity = \cos(\theta) = \frac{x * y}{\|x\|\|y\|} = \frac{\sum_{i=1}^{n} x_i\, y_i}{\sqrt{\sum_{i=1}^{n} x_i^2}\ \sqrt{\sum_{i=1}^{n} y_i^2}}$$

Cosines range from $-1$ (opposite) to 1 (exact or same). However, the cosine similarity or match score between two bodies of text (i.e., an expectation and a student response) in this context functionally ranges from 0 to 1 as a result of TF-IDF weighting where TF-IDF matrix term frequencies will never be negative.

### 1.3. Corpora for Training Semantic Models

AutoTutor frequently has created LSA spaces that use the TASA corpus (Touchstone Applied Science Associates, Inc., Brewster, NY, USA) [25]. TASA was created by systematically sampling documents from a variety of news articles, novels, and other texts to create a corpus of relevant words and documents that individuals are exposed to in different grades, from K to college-ready. There is an attempt to have a comparable number of texts at each grade level so that the difficulty of texts can be computed in addition to the genre of texts (e.g., expository, narrative). The TASA space is regarded as a general English language LSA space. For this research, rather than using the popular TASA LSA space, an Electronics LSA space was developed by creating a corpus of texts from electronics manuals and relevant curriculum materials in addition to a previously constructed corpus of 32,000 Newtonian physics documents that are relevant to electronics. A semantic similarity model trained on a specific subject (i.e., LSA spaces trained on electronics for the purpose of grading student quiz answers in an electronics class) is expected to assess student input in domain-specific ASAG tasks more accurately and minimize errors attributable to the use of general English language terms. These models can be seen as semantic representations of the text corpora used in their training.

In this study, we confirmed that the best performance was in models trained on an electronics and physics corpus. An electronics corpus was developed in 2021 by university researchers using a collection of electrical engineering textbooks, manuals, lecture notes, and the Navy Electricity and Electronics Training Series. The corpus had approximately 10 million subject-specific word or symbol tokens. Physics texts were included in the training corpus because physics and electrical engineering may share some overlap in subject matter, technical language, and application. Some subfields of physics interface with subfields of engineering (e.g., the relationship between optics/photonics and electrical engineering). Additionally, physics texts appear to slightly improve model performance on semantic matches.

### 1.4. Comparing Distributional Semantics, Word Embeddings, and Contextual Embeddings

Before explaining the differences between these types of models, it is important to note four types of training that produce them. Four typical forms of training for language models are unsupervised, supervised, self-supervised, and semi-supervised. *Unsupervised learning* is a type of machine learning wherein the algorithm learns patterns and structures in data without any labeled examples or explicit guidance. In *supervised learning*, a machine

learning algorithm learns from labeled examples or training data to make predictions or classify unseen data using input features and corresponding target labels.

*Self-supervised learning* is a form of unsupervised learning wherein the algorithm learns to predict missing or corrupted parts of the data itself. *Semi-supervised learning* is a combination of supervised and unsupervised learning techniques. It leverages a small amount of labeled data along with a larger amount of unlabeled data to improve the learning process.

Word embedding models (e.g., Word2Vec) are more modern than distributional semantics techniques such as LSA for computing semantic similarity. While they are both used to represent words in a corpus, the approaches they take are different. Unlike LSA, Word2Vec is a neural network algorithm that also represents words in a vector space model. In contrast to the statistical approach used by LSA, it uses a shallow, two-layer neural network to learn the vector representations of words. LSA is better suited for analyzing the semantic relationships between words in a document based on a corpus, whereas Word2Vec is better suited for capturing the relationships between words from the sentence context in which they appear. This is because Word2Vec considers words surrounding the target word within a fixed context window (e.g., a five-word window where the target word is always the third or middle word) and LSA does not.

This notion of context can be confusing here. For both Word2Vec and BERT, word embeddings at the input layer are inherently non-contextual from the standpoint of documents but are contextualized when hidden layers are used to extract the meaning of a given word, depending on the words surrounding it in a sentence. Using one hidden layer, Word2Vec learns relationships between words by analyzing word context within a sentence or context window, but not between sentences in a document. A transformer-based model (i.e., contextual embeddings) like BERT or GPT also captures individual word context in sentences as well as between sentences within documents but uses several hidden layers. In both Word2Vec and BERT variants, hidden layers do extract meaning from surrounding words and sentences, whereas the embeddings at the input layer do not.

Unlike Word2Vec, BERT uses a deep bidirectional transformer with 12 hidden layers to learn the vector representations of words. The use of deep bidirectional transformers paired with Siamese [26] and triplet network structures [27] is what allows transformer models such as BERT to handle context more completely and form more nuanced semantic and syntactic relationships than Word2Vec.

LSA may perform perfectly well for tasks like ASAG on such technical topics as electronics. However, there may be improvements in performance with these more recent models. Moreover, the relative performance of prediction-based methods compared to count-based methods is unclear for automatic grading tasks [28,29]. Consequently, the goal of this paper is to evaluate the LSA model used in ElectronixTutor and compare the performance of this model to more contemporary word embedding models like Word2Vec variants, as well as contextual embeddings using state-of-the-art transformer models such as SBERT and SGPT. In addition to evaluating the performance of each of the stand-alone models mentioned, this paper observes each model combined with the RegEx model.

### 1.5. Word2Vec Variants and Training

Word2Vec and its variants can be substituted for LSA in applications such as assessing semantic textual similarity of student verbal input against an expected answer in ASAG. Word2Vec creates semantic field representations, typically using one of two techniques, Continuous Bag of Words (*CBOW*) or Skip-gram. In the CBOW model, distributed contextual representations are used to predict a word, whereas in Skip-gram, distributed word representations attempt to predict the context.

Bigram2Vec (*W2VB*) is another variation of Word2Vec observed in the study. This is a variation of the same Word2Vec model that focuses on bigram embeddings rather than single-word embeddings. Doc2Vec (*D2V*), the last Word2Vec variant observed in the study, is computed by taking a Word2Vec model and adding one simple parameter, a paragraph

or document ID. Word2Vec and variants, like LSA, have the ability to compute cosine similarity scores between two vectorized text inputs, serving as a basis for comparison between semantic models on the task.

Word2Vec models have previously been compared to distributional semantics techniques, such as LSA, using cosine similarity metrics [30,31]. Word2Vec models may outperform LSA models for text similarity tasks in particular contexts. In the case of LSA, the degree to which differing models can be fairly compared in similar tasks depends on the training corpus, among other factors, such as domain specificity, dimensionality, and weighting. All Word2Vec variants are trained on the same corpora used to train LSA models.

### 1.6. BERT Variants and Training

In addition to substituting Word2Vec models for LSA in automatic grading tasks, we included newer state-of-the-art models. Two such models are BERT and SBERT [21,22]. The original BERT model was pre-trained on a large corpus of 3.3 billion words. The SBERT model tested in this paper is also trained on the Stanford Natural Language Inference and Multi-Genre Natural Language Inference corpora [32,33]. In contrast, Word2Vec variants and LSA were trained in this task as unsupervised models on a 10-million-token corpus specified for electronics and physics.

In BERT variants, downstream tasks and continued pre-training for domain adaptation are considered self-supervised. That is, training on a very large corpus of texts involves many cycles in which (a) artificially labeled (surrogate supervised) predictions are made on each word in a very large corpus, based on other words around the word, (b) the observed word is received, and (c) the statistical knowledge representation is adjusted through continued pre-training when there are discrepancies between the predicted and observed word.

### 1.7. GPT-3 Variants and Training

Another current model that has achieved benchmark success in several relevant NLP tasks is GPT-3 by OpenAI [23]. Whereas BERT uses 110 million parameters with 12 hidden layers, GPT-3 uses 175 billion parameters and 96 hidden layers. While both SBERT and GPT-3 are based on the same transformer architecture, SBERT is specifically designed for sentence embedding tasks, whereas GPT-3 is designed for natural language generation tasks. For example, one variant of GPT-3 that has gained recent international popularity in research and on social media is ChatGPT. This widely popular model allows users to hold open conversation with prompting and question asking across many domains and has become the gold standard for sentence and essay generation.

While intended use cases differ between GPT-3 and BERT variants, we use a variant of GPT-3 by Muennighoff, 2022 [24] called Sentence-GPT (or *SGPT*) that is suited to encode sentences for semantic similarity in the task analyzed in the present study. SGPT can be configured for either symmetric or asymmetric tasks. At 5.8 billion parameters, the SGPT bi-encoder produces state-of-the-art natural language sentence embeddings for the task of semantic search [24]. The SGPT model in the study uses a unique position-weighted pooling method and isolates only the bias vectors for fine-tuning. This closely follows recent work for performing supervised contrastive learning [34,35] with in-batch negatives in order to optimize the cost function (i.e., negative examples that are sampled from within the same batch during training that allow the model to learn from comparisons made within the batch). Specifically, the mean pooling method used in SGPT gives later tokens a higher weight than a standard mean pooling method.

### 1.8. Pre-Trained Transformers in ASAG Tasks for Conversational Tutoring

SBERT produced favorable performance over BERT in several tasks including semantic text similarity for sentence embeddings [22]. However, this was achieved through a considerably high degree of supervision. Accordingly, SBERT, as well as other transformer

models [36] may remain ineffective and unreliable in contexts such as ASAG for a domain-specific topic such as electronics. In such topics, large, labeled data may be scarce [37] and many relevant terms in the materials are distinct to the subject matter of electronics rather than general English language. The literature favors unsupervised approaches in these tasks [38]. Fine-tuning may improve domain specificity for large pre-trained transformers (both BERT and GPT-3 variants) but requires a new, large dataset for each task and may limit the model's ability to generalize out of the training corpus [39].

Another constraint of using large, pre-trained language models such as BERT is that they are not sensitive to the compositional constraints of ElectronixTutor that decompose good answers into expectations and words or phrases within expectations. This is critical for designing a conversation with the computer administering feedback, hints, and prompts to guide the students' verbal contributions. Supervised methods can eventually be used to improve the performance of semantic models in conversational ITSs when a dataset is collected from thousands of learners.

## 2. The Current Study

In this paper, we analyzed the performance of LSA, Word2Vec, variants of Word2Vec, SBERT, and SGPT using performance metrics routinely collected in computational linguistics and social sciences. We computed metrics again for each of these model types combined with RegEx. Four performance measures routinely collected in the field of computational linguistics were computed: precision, recall, accuracy, and F1 (i.e., the harmonic mean of precision and recall). Cohen's kappa ($\kappa$) was also calculated for each model, a metric that is widely accepted in the social sciences. For assessing performance across models, we focused mainly on F1 and $\kappa$, where the F1 score can be thought of as the weighted measure between precision and recall. This is a simple, balanced reference point that considers both precision and recall. These performance metrics were collected to compare agreement between the computer models and human judges as well as between human judges.

A comparison of two judgments of similarity has four possible outcomes that can be specified in a simple matrix. These four possible outcomes are true positives (TP, often called hits), true negatives (TN, often called correct rejections), false positives (FP, often called false alarms), and false negatives (FN, often called misses). Using these four outcomes, we compute $recall = \frac{TP}{TP+FN}$, $precision = \frac{TP}{TP+FP}$, $accuracy = \frac{TP+TN}{TP+TN+FP+FN}$, and $F1 = \frac{2*TP}{(2*TP)+FP+FN}$ along with Cohen's kappa ($\kappa$). Cohen's kappa is a measure of agreement for categorical items between multiple raters (e.g., two humans or a human and a computer grading model). Cohen's kappa is an unbiased measure, unlike percent agreement [40], because it adjusts for the posteriori base rates of positive versus negative decisions.

## 3. Method

### 3.1. Model Comparison and Corpus Specification

We assessed stand-alone distributional semantics and RegEx models as well as semantics and RegEx combination models. The stand-alone models for assessing student verbal contributions were assessed for performance and compared to other recent stand-alone models as well as the original LSA model reported by Carmon et al., 2019 [41]. This study assessed the performance of 13 models: LSA, W2V, W2VB, D2V, SBERT, SGPT, and RegEx stand-alone models, as well as combination models for each of the corpus-based stand-alone models paired with RegEx performance.

The LSA, W2V, W2VB, and D2V models were trained on the same corpus of 32,000 Newtonian physics texts for the first set of results in the study; they were subsequently trained on a hybrid corpus of Newtonian physics texts and a large collection of electronics texts ranging from electronics manuals and textbooks on specialized electronics topics to curriculum notes for electronics courses. In contrast, SBERT and SGPT models used very large pre-trained corpora available from the SNLI and Multi-Genre NLI sites. Semantic models trained on different corpora are incommensurate for comparison, but we include SBERT

and SGPT here to highlight the differences between large, pre-trained general language models and models trained on domain-specific corpora for the given task.

### 3.2. Data Collection

We analyzed student response data from initial answers to questions used in a conversational ElectronixTutor ITS [41,42] in which students answer electronics questions. The responses were scored by the computer (the 13 models) and then compared to categorical ratings of human judges on a scale that varies from 1 to 6. We collected the dataset on 219 unique Amazon Mechanical Turk workers (MTurkers) who collectively answered 118 questions asked by the AutoTutor component of ElectronixTutor. MTurkers responded to these questions in an open-ended fashion, typing in as much or as little as needed in order to fully answer the questions asked. Each question received up to 20 user responses, yielding 2350 responses to the 118 main questions. Each MTurker worker answered a small subset of the 118 main questions. Of the 2350 collected responses to the main question, each response was paired with the ideal answer and each expectation associated with the main question, resulting in 5166 ($n$ = 5166) total response pairings that were used as the sample units for the analyses.

### 3.3. Human Judges

The student responses were rated by two subject-matter experts on electronics, independently. The electronics subject-matter experts received a training session on evaluating semantic similarity between ideal answers to main questions, expectations, and participant responses. Subject matter experts had graduate degrees in electronics, with one serving as an instructor in engineering classes. Judges were trained on a small number of the MTurker responses to questions and were asked to share ratings. The judges then discussed and justified their ratings for the training items amongst each other. The response rating procedure included a judge rating tool to simplify the rating process, save time, and reduce the cognitive load taken on by the judges. The rating tool displayed the question or part of the question applicable, the participant response, the expectation, and a field for specifying a 1–6 rating. The following scale was used by human experts in evaluating answers:

(1) There was no attempt to answer the question.
(2) The answer is not on topic or contains metacognitive language.
(3) The answer is on topic, but completely incorrect.
(4) The answer is mostly incorrect but contains a small degree of truth value.
(5) The answer is mostly correct.
(6) The answer is ideal.

The following is an example of student answers to the example main question (presented in the Introduction) that received ratings of 6, 5, and 4 by human judges, respectively:

*Student Initial Answer Rated 6:* The voltage at the bulb will stay at the maximum of the Zener voltage. This way any excess current that may have resulted from overvoltage and burned out the bulb will be drained by the Zener diode. If the voltage at the bulb goes beyond the Zener breakdown voltage the diode will begin to conduct and pass as much current as is necessary to keep the voltage at the bulb constant.
*Student Initial Answer Rated 5:* The diode provides an alternate path for current to flow and it prevents some of the current from flowing in the wrong direction. In this way it protects the bulb from surges better than if the bulb were subjected to all the current by itself.
*Student Initial Answer Rated 4:* It diverts the energy elsewhere into a place where it can gather and settle to keep the bulb from overheating too fast if too much power collects inside of the bulb from bursting.

### 3.4. Coding for Analyses

We analyzed the extent to which the answer responses were rated as ideal by human judges (i.e., student responses that judges rated as 6) and compared them with automated

models that scored answers with cosine similarity optimized for stringency (as defined shortly). We examined the computer's ability to agree with student responses that were rated as a 6 by the human judges because accurately identifying ideal answers to the main question is important for subsequent branching of dialogue moves in conversational ITSs. A stringent criterion is also adopted in the Navy for advancement in courses, as discussed earlier. We did not analyze more lenient criteria for matches in the present study.

To obtain performance metrics for model agreement with humans, raw scores from computer models were coded to 1s and 0s in comparison to human judge ratings of 6. A score of 1 for computer models is contingent on whether the discrimination threshold for the model is passed. In the case of the distributional semantics and word embedding models in this paper, the discrimination threshold is based on cosine similarity values.

### 3.5. Optimizing F1 between Human and Computer Scoring

We optimized the cosine similarity threshold by simulating agreement for all possible threshold values for computer scoring and identifying the highest point of agreement. Previously, the literature has addressed the notion of optimizing cosine similarity threshold values for performance [43].

To identify the appropriate discrimination threshold for each model observed in the study, we wrote a model simulation in Python. The simulation computes the discrimination threshold that optimizes F1 values between the computer and humans by simulating each value of a computer model's discrimination threshold from 0 to 1 in thousandths (0.001, 0.002, 0.003, . . ., 0.999) and outputting 1000 sets of performance scores. This allows us to identify the optimal F1 value and plot the output. This simulation was conducted on stand-alone semantic models separately from the RegEx and semantics combination models.

Figure 2 shows an example of simulation outputs plotted for the Word2Vec stand-alone models and combination model. The Word2Vec optimal threshold value is approximately 0.8 for the stand-alone model and 0.9 for the combination model.

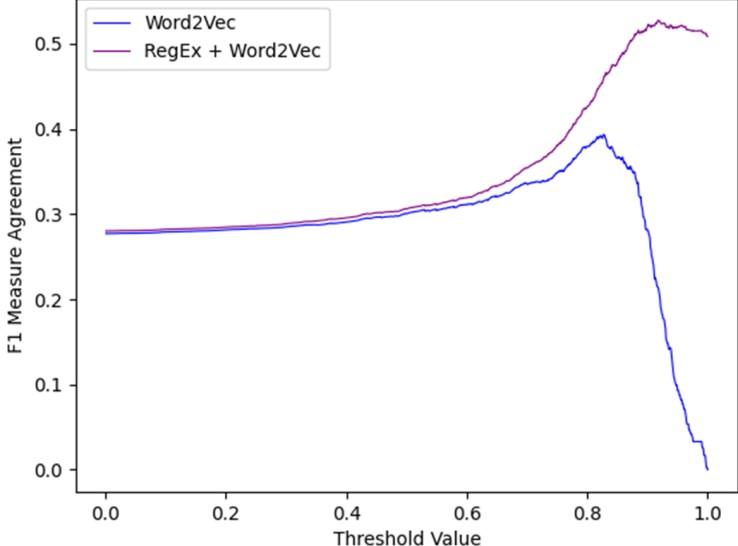

**Figure 2.** F1 measures simulated and plotted for 1000 threshold values in stand-alone Word2Vec and combination Word2Vec models.

Table 1 specifies optimal thresholds identified by simulating the computer models for human–computer agreement in computer (*c*) and human (*h*), respectively, where *T* is threshold, and human rating thresholds are constant, $T_h = 1$, *rating* > 5; *else*, $T_h = 0$.

**Table 1.** Thresholds in computer models for computing optimal F1.

| Model | Observed Threshold |
|---|---|
| RE | $T_c = 1$, $T_{RE} \geq 0.750$; *else*, $T_c = 0$ |
| LSA | $T_c = 1$, $T_{LSA} \geq 0.510$; *else*, $T_c = 0$ |
| W2V | $T_c = 1$, $T_{W2V} \geq 0.819$ ; *else*, $T_c = 0$ |
| W2VB | $T_c = 1$, $T_{W2VB} \geq 0.839$; *else*, $T_c = 0$ |
| D2V | $T_c = 1$, $T_{D2V} \geq 0.839$; *else*, $T_c = 0$ |
| SBERT | $T_c = 1$, $T_{SBERT} \geq 0.523$; *else*, $T_c = 0$ |
| SGPT | $T_c = 1$, $T_{SGPT} \geq 0.647$; *else*, $T_c = 0$ |
| RE + LSA | $T_c = 1$, $T_{LSA} \geq 0.915 \vee T_{RE} \geq 0.750$; *else*, $T_c = 0$ |
| RE + W2V | $T_c = 1$, $T_{W2V} \geq 0.951 \vee T_{RE} \geq 0.750$; *else*, $T_c = 0$ |
| RE + W2VB | $T_c = 1$, $T_{W2VB} \geq 0.929 \vee T_{RE} \geq 0.750$; *else*, $T_c = 0$ |
| RE + D2V | $T_c = 1$, $T_{D2V} \geq 0.915 \vee T_{RE} \geq 0.750$; *else*, $T_c = 0$ |
| RE + SBERT | $T_c = 1$, $T_{SBERT} \geq 0.770 \vee T_{RE} \geq 0.750$; *else*, $T_c = 0$ |
| RE + SGPT | $T_c = 1$, $T_{SGPT} \geq 0.860 \vee T_{RE} \geq 0.750$; *else*, $T_c = 0$ |

## 4. Results and Discussion

We examined the degree to which the LSA model used in ElectronixTutor compared to the performances of other semantic text similarity models. Performance on these models is interpreted in comparison to agreement between the two experts. The agreement between two human experts showed a precision score of 0.467, a recall score of 0.618, accuracy of 0.867, an F1 measure of 0.532, and a $\kappa$ score of 0.456. These scores indicate there is moderate but not perfect agreement among human experts. Computer models would show impressive performance if their comparisons with human experts were equal to the agreement between human experts.

### 4.1. Agreement between Humans and Computer Models

Tables 2 and 3 show results for each stand-alone model compared with the two human experts. Table 2 shows results when the Newtonian physics LSA space was used, whereas Table 3 shows results when the models (except for SBERT and SGPT) were trained on the larger corpus that includes both electronics and Newtonian physics documents. *RE* refers to Regular Expressions, *W2V* is Word2Vec, *W2VB* is Bigram2Vec, *D2V* is Doc2Vec, *SBERT* is SBERT, and *SGPT* is Sentence-GPT. Table 4 shows results for the corpus-based models when combined with the RE model.

**Table 2.** Agreement between Judge 1 and Judge 2 versus stand-alone computer model. There are models with a domain-specific corpus (Newtonian physics minus electronics) and very large pre-trained corpora (SBERT and SGPT).

| Performance Metrics | | Judge 1 | | | | | |
|---|---|---|---|---|---|---|---|
| | LSAp | W2V | W2VB | D2V | RE | SBERT | SGPT |
| Precision | 0.644 | 0.492 | 0.583 | 0.469 | 0.448 | 0.571 | 0.475 |
| Recall | 0.242 | 0.284 | 0.243 | 0.276 | 0.440 | 0.277 | 0.284 |
| Accuracy | 0.610 | 0.718 | 0.640 | 0.715 | 0.794 | 0.690 | 0.722 |
| F1 | 0.351 | 0.360 | 0.343 | 0.347 | 0.444 | 0.373 | 0.356 |
| $\kappa$ | 0.153 | 0.196 | 0.150 | 0.181 | 0.366 | 0.200 | 0.193 |
| | | Judge 2 | | | | | |
| Precision | 0.640 | 0.450 | 0.562 | 0.429 | 0.450 | 0.510 | 0.429 |
| Recall | 0.180 | 0.205 | 0.243 | 0.196 | 0.585 | 0.187 | 0.195 |
| Accuracy | 0.602 | 0.720 | 0.617 | 0.715 | 0.794 | 0.670 | 0.713 |
| F1 | 0.281 | 0.282 | 0.262 | 0.269 | 0.509 | 0.274 | 0.268 |
| $\kappa$ | 0.113 | 0.137 | 0.095 | 0.122 | 0.428 | 0.116 | 0.120 |

**Table 3.** Agreement between Judge 1 and 2 versus stand-alone computer model. There are models with a domain-specific corpus (Newtonian physics plus electronics) and very large pre-trained corpora (SBERT and SGPT).

| Performance Metrics | Judge 1 | | | | | | |
|---|---|---|---|---|---|---|---|
| | LSAc | W2V | W2VB | D2V | RE | SBERT | SGPT |
| Precision | 0.583 | 0.596 | 0.560 | 0.539 | 0.448 | 0.571 | 0.475 |
| Recall | 0.272 | 0.292 | 0.250 | 0.297 | 0.440 | 0.277 | 0.284 |
| Accuracy | 0.680 | 0.707 | 0.660 | 0.720 | 0.794 | 0.690 | 0.722 |
| F1 | 0.370 | 0.386 | 0.347 | 0.383 | 0.444 | 0.373 | 0.356 |
| $K$ | 0.194 | 0.220 | 0.160 | 0.221 | 0.366 | 0.200 | 0.193 |
| | Judge 2 | | | | | | |
| Precision | 0.577 | 0.521 | 0.526 | 0.481 | 0.450 | 0.510 | 0.429 |
| Recall | 0.201 | 0.204 | 0.176 | 0.203 | 0.585 | 0.187 | 0.195 |
| Accuracy | 0.670 | 0.690 | 0.643 | 0.706 | 0.794 | 0.670 | 0.713 |
| F1 | 0.398 | 0.293 | 0.264 | 0.286 | 0.509 | 0.274 | 0.268 |
| $K$ | 0.144 | 0.142 | 0.100 | 0.138 | 0.428 | 0.116 | 0.120 |

**Table 4.** Agreement between Judge 1 and 2 versus combination computer model. There are models with a domain-specific corpus (Newtonian physics plus electronics) and very large pre-trained corpora (SBERT and SGPT), both combined with RE.

| Performance Metrics | Judge 1 | | | | | |
|---|---|---|---|---|---|---|
| | RE + LSAc | RE + W2V | RE + W2VB | RE + D2V | RE + SBERT | RE + SGPT |
| Precision | 0.479 | 0.502 | 0.450 | 0.462 | 0.450 | 0.473 |
| Recall | 0.584 | 0.562 | 0.585 | 0.587 | 0.584 | 0.577 |
| Accuracy | 0.852 | 0.856 | 0.859 | 0.860 | 0.859 | 0.859 |
| F1 | 0.526 | 0.530 | 0.508 | 0.517 | 0.517 | 0.519 |
| $K$ | 0.446 | 0.446 | 0.428 | 0.437 | 0.366 | 0.438 |
| | Judge 2 | | | | | |
| Precision | 0.482 | 0.529 | 0.496 | 0.458 | 0.448 | 0.468 |
| Recall | 0.440 | 0.405 | 0.349 | 0.440 | 0.440 | 0.431 |
| Accuracy | 0.851 | 0.847 | 0.825 | 0.862 | 0.862 | 0.860 |
| F1 | 0.458 | 0.460 | 0.410 | 0.449 | 0.444 | 0.448 |
| $K$ | 0.371 | 0.372 | 0.312 | 0.312 | 0.366 | 0.368 |

### 4.2. Major Trends in the Patterns of Scores

The mean scores in Tables 2–4 support a number of general observations. First, the Regular Expressions (RE) model clearly reigns supreme in comparisons with human expert ratings when adopting a stringent criterion in semantic matches. RE, in the stand-alone model comparisons, had an F1 score of 0.509, which is almost as high as the 0.532 F1 agreement between human experts. The highest stand-alone corpus-based model was 0.398 for LSA when trained on the physics and electronics corpus. RE had higher scores in 24 out of the 24 comparisons of means, which is statistically significant according to a nonparametric Wilcoxon sign test ($p < 0.001$).

Second, LSA fared extremely well when compared with the other corpus-based models. When averaging scores for the two judges and two domain-specific corpora, the mean F1 score for the LSA model was 0.350, which compares very well with the highest-performing other stand-alone corpus-based model, namely Word2Vec (0.330). Of the 20 comparisons between LSA and the other models, 14 were in favor of LSA, which is statistically significant according to a sign test. When considering the combined models with RE, the LSA model fared quite well (0.492) compared with the other best-performing corpus-based model, namely Word2Vec (0.495). Eight of the ten comparisons favored LSA. Consequently, the more recent distributional semantic models do not eclipse the performance of LSA in this study in the electronics domain. If anything, LSA was the best model behind Word2Vec, which performed better by a slight margin.

Third, there are some interesting trends regarding the corpora for training the corpus-based models. The mean of the stand-alone models with the smaller domain-specific

corpora (LSA, W2V, W2VB, and D2V) was 0.326, which is a bit higher than the 0.318 value of the extremely large pre-trained models with domain-general corpora (SBERT and SGPT); however, the differences were not significant according to a sign test. Similar results were found when examining the combined models with RE. However, the domain-specificity of the corpus needs to be considered as well as the size of the corpus. The large corpus models were domain general, whereas the small corpus models were domain specific. The corpus that included both physics and electronics texts (an advantage of both size and domain specificity) did have higher F1 scores than the physics-only corpus when means are compared in Table 2 versus Table 3 for LSA, W2V, W2VB, and D2V models; 0.341 versus 0.269, respectively; eight out of eight comparisons favored the larger domain-specific corpus, which is significant according a sign test ($p < 0.001$). However, it is not possible to determine whether it is size or domain specificity that explains the difference.

Fourth, the corpus-based models were robustly enhanced by including the RE compared with the stand-alone models, with means of 0.482 and 0.333, respectively; 12 out of 12 comparisons were in the expected direction, which is significant according to a sign test ($p < 0.001$). So, the combined models show enhancements when RegExes are added. Very unexpectedly, however, the RE model alone was almost as high as the mean of the combined models (0.482), which raises the question of the added value of the corpus-based models over the RegEx modules alone. The combined model of W2V with RE was indeed higher than RE alone (0.495 versus 0.477) but parametric statistical analyses would need to be conducted to assess whether the difference is statistically significant. The combined model of LSA and RE was also higher in magnitude than RE alone; again, parametric statistics would need to assess whether this difference is significant, statistically. Nevertheless, the scores are in the right direction and confirm the added value of combined LSA and RE models over LSA-alone and RE-alone models.

### 4.3. Performance of Word and Contextual Embeddings

W2V variants trained on the physics corpus performed very similarly to one another, excluding Bigram2Vec. Performance in the electronics-trained stand-alone models was similar to but better than model performance in the physics-trained condition. In the electronics-trained stand-alone models, each model performed better overall than it previously did when trained on the physics-only corpus.

According to the results, W2V showed promise as having the highest F1 and $\kappa$ values in the task, and had accuracy advantages over the LSA space from 2019, with accuracy reaching 0.852 averaged between the judges in the RegEx and W2V combination model. W2V typically beat SBERT and SGPT in stand-alone conditions when W2V was trained on an electronics corpus as well as in RegEx combination computer models.

The Bigram2Vec model W2VB had lower but similar scores in metrics like accuracy and F1, but unfortunately performed the worst out of any observed model in the study in physics-trained, electronics-trained, and combination model conditions. W2VB had the highest precision in stand-alone models, but considering the other measures and preferences for recall over precision in ASAG systems, that was not such a good thing on its own. Bigram2Vec had much lower accuracy than all other stand-alone models in the study, and it was more than just a marginal difference in accuracy. The model was ultimately too lenient in awarding high match scores between unlike terms.

The D2V model had higher precision and F1 than both the original and replicated LSA space while retaining similar recall, accuracy, and $\kappa$. This model performed consistently well in physics-trained, electronics-trained, and combination model conditions. The D2V model was typically outperformed in this study by the LSA and W2V models. While the model was typically outperformed in overall performance measured by F1 and $\kappa$, it was very similar to LSA and W2V (the highest-performing models in the study). Further, D2V models in this study typically had higher accuracy and recall than other models observed in the same condition, which is worth noting, because in conversational ITSs, which perform assessments of verbal input through ASAG techniques, recall appears to be valued higher

than precision. That is because we would rather pass a student answer that is slightly wrong than discount a fully correct student answer, which may discourage the student from continuing engagement with the system.

Lastly, SBERT and SGPT performed worse than RegEx in stand-alone conditions, but slightly better than other semantic models when they were only trained on Newtonian physics. This comparison may be misleading, as all semantic models were trained on the same corpus except the pre-trained SBERT and SGPT models. Unsupervised LSA or W2V variants trained on electronics typically outperformed pre-trained SBERT and SGPT by a slight degree on most relevant metrics, including $\kappa$ and F1. This difference is most easily attributed to the domain-specific corpus used in unsupervised training. This advantage is also realized when considering the lower performance for unsupervised models on physics-only corpora when compared to a physics plus electronics corpus on the task of automatically grading electronics answers. It is unclear the degree to which fine-tuning a large pre-trained transformer for electronics would affect the model's performance.

## 5. General Discussion

### 5.1. Stand-Alone Models and Combination Computer Models

The scores in the stand-alone models are a bit low, but this was to be expected. This observation may be misleading without proper context. A combination of RegEx and LSA makes for a more appropriate model than using distributional semantics techniques alone (such as a stand-alone LSA model). This was indeed the case. In the current study, which used combinations of RegEx and semantic models to assess student verbal input, agreement was low between the judges and stand-alone semantic models (LSA, W2V, W2VB, D2V, SBERT, and SGPT) and a bit higher between the judges and the RegEx model. It is only when RegEx is combined with a semantic model and compared to humans that we start to see higher agreement. The current study showed low agreement between semantic models and humans, higher agreement between RegEx and humans, and then combination models that agree more with humans than any of the stand-alone RegEx or semantic text models. Combination models with semantic models trained on the electronics plus physics corpus tended to agree with human judges more closely in the study than combination models trained on physics only.

### 5.2. Interpretation of Human and Computer Agreement on Task Data

The agreement between human judges was not considered to be extremely high in this study, but this is not unlike other studies that report Cohen's kappa and F-measures [44]. Studies that report percent agreement and correlations frequently have limitations of high a posteriori base rates, observation selection bias, and distribution characteristics that run the risk of inflating the magnitude of the similarity indices. A common interpretation bias for inter-rater agreement assumes that high inter-rater agreement entails high-quality data. This is not always the case. The important thing is that computer–human agreement is moderate and similar to agreement between humans. In fact, lower inter-rater agreement may suggest natural subjectivity in human judgment, especially in complex knowledge domains. Further, inter-rater agreement is typically lower for a fine-grained 6-point scale (e.g., text relevance scales) than for binary decisions [45].

### 5.3. Considerations for Structuring and Collecting New Data for Further Analysis

We may see considerably higher F1 scores by coding the 6-point scale to a 4-point scale, or by observing one-off agreements (range of agreement) rather than precise agreements on a 6-point scale. Additionally, lowering the threshold from observing judge ratings of 6 will result in much higher F1 scores, but this changes the implications of this analysis in detecting ideal answers to the main question for accurately branching to the most appropriate next dialogue move.

In 2022, we collected new log data on Navy trainees who respond to electronics questions. The quality of the data and frequency of null-value student responses in the record

store appear promising, at a glance. This may, in part, mitigate the large representation of true negatives observed in the current dataset. The newer data contain a greater number of parameters for meta-data which will allow us to conduct analyses on more specific questions with a finer grain of detail.

### 5.4. Conclusions and Implications

The highest F1 score observed in combination models came from the RegEx and W2V combination, reaching 0.530. This is perhaps of the most notable findings in the paper, considering that agreement between humans for ideal answers measured a similar F1 score of 0.532, $\kappa = 0.456$, while accuracy between humans was 0.867, and accuracy reached 0.856 in the RegEx and W2V combination model. W2V also performed better than large pre-trained transformer models for general language. This indicates the importance of domain-specific training corpora when specifying semantic models for such a task, especially in combination with RegEx.

Large pre-trained transformer models (SBERT and SGPT) may be appropriate for certain tasks, but not the current one, despite achieving state-of-the-art performance on general-language-adjacent tasks and out-of-sample question tasks [46]. Notably, comparisons on models trained with different corpora on domain-specific ASAG tasks should be avoided when possible. These comparisons are incommensurable, as the text corpora used to train compared models may substantially differ in subject matter; therefore, each model yields different semantic representations. Furthermore, we see that unsupervised methods outperform large, pre-trained, self-supervised methods when the training corpus is more specific to the domain of the task, electronics.

The W2V model performed better overall than other models in the physics-trained condition, including LSA, SBERT, and SGPT. Additionally, by acquiring more specific electronics texts to add to the corpus for model training, we observed a substantial increase in the performance of each model observed in the study, especially in unsupervised word embedding models in the task, such as the W2V and D2V models. Overall, the results here are very encouraging. In combination models, performance was much higher than in stand-alone models. RegExes, when written robustly, appear to handle domain-specific ASAG much better than semantic text models in stand-alone model conditions. The crucial takeaway here is that in pairing both RegEx and semantic text models into a single combination model, we see clearly higher agreement with humans than in stand-alone RegEx and semantic text models.

### 5.5. Future Research

We should consider removing the Bigram2Vec condition and replacing it with other similar and newer word embedding and transformer techniques. Additionally, it is necessary to address dimensionality and corpus size effects on model performance in this context. As previously noted, Altszyler et al., 2016 [30] suggested performance similarity in LSA and W2V Skip-gram models. W2V Skip-gram starts to perform better than LSA, and the difference in performance becomes more pronounced as corpus size grows and the token count increases towards 10 million words. Incidentally, the combined corpus of both physics and electronics used in the current study trained models on approximately 10 million words. Perhaps increasing corpus size may produce favorable performance increases in the RegEx and W2V model. Additionally, modifying content, addressing dimensionality, and exploring alternative unsupervised and self-supervised embedding models in combination with RegEx is of interest in informing the next steps in future research.

While SBERT models with large, supervised pre-training corpora may be impractical on this task [37–39], a critical next step in this work emerges as the need to pass our electronics plus physics corpus through transformer models for fine-tuning. Perhaps fine-tuning for domain-specific electronics corpora will increase the performance of SBERT and SGPT on this task to make performance better or more similar to other unsupervised models

trained on electronics corpora in the study. We should run current, newer, and adjusted training models (e.g., fine-tuned pre-trained transformers) on new data that have been collected from Navy electrical engineers in 2022. There were semantic judgments between student responses and expectations from the three human experts in this recent study. There were also many student contributions in the multi-turn follow-up conversations after the initial response of the human from the main question. How will the different models compare in evaluating student contributions across many turns in the AutoTutor conversations?

**Author Contributions:** Conceptualization, C.M.C., B.M., X.H. and A.C.G.; methodology, C.M.C., B.M. and A.C.G.; software, C.M.C.; validation, C.M.C. and A.C.G.; formal analysis, C.M.C. and A.C.G.; investigation, C.M.C. and A.C.G.; resources, C.M.C., B.M. and X.H.; data curation, C.M.C. and B.M.; writing—original draft preparation, C.M.C., B.M. and A.C.G.; writing—review and editing, C.M.C. and A.C.G.; visualization, C.M.C.; supervision, C.M.C., B.M., X.H. and A.C.G.; project administration, X.H. and A.C.G.; funding acquisition, X.H. and A.C.G. All authors have read and agreed to the published version of the manuscript.

**Funding:** This research was supported by the Office of Naval Research (N00014-00-1-0600, N00014-15-P-1184; N00014-12-C-0643; N00014-16-C-3027) and the National Science Foundation Data Infrastructure Building Blocks program (ACI-1443068). Any opinions, findings, and conclusions or recommendations expressed in this material are those of the authors and do not necessarily reflect the views of ONR or NSF.

**Data Availability Statement:** The response data and training corpora for semantic models are available upon request from corresponding author, C. M. Carmon.

**Conflicts of Interest:** The authors declare no conflict of interest.

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
