# Peer review of "Automated Assessment of Initial Answers to Questions in Conversational Intelligent Tutoring Systems: Are Contextual Embedding Models Really Better?"

_electronics, doi:10.3390/electronics12173654_

Round 1
Reviewer 1 Report
The paper aims to evaluate the effectiveness of semantic text models in grading student responses in electronics compared to human judges. Despite the availability of various text similarity models, it remains to be seen if these newer models perform as well as earlier distributional semantics models.
Findings:
Among the standalone models, regular words achieved the best performance. However, other semantic text models demonstrated comparable results to the Latent Semantic Analysis sample initially used for this task and, in some cases, even outperformed it. Notably, models trained on domain-specific electronics corpus performed better than those trained on general language or Newtonian physics. Incorporating semantic text models with regular expressions yielded better results than standalone measures, which aligned well with human judges' assessments.
Implications:
These findings highlight the importance of empirical analysis in selecting appropriate computational procedures for text modeling, particularly in domain-specific areas like electronics. The study improves the accuracy, recall, weighted agreement, and overall effectiveness of automatic scoring in everyday Intelligent Tutoring Systems. It also raises the question of how recent contextual embedding models compare with earlier distributional semantic language models. I recommend to add in your background the following work https://doi.org/10.3390/app13020677
The study demonstrates the potential of various semantic text models in grading electronic student responses. It shows that while regular expressions performed the best among standalone models, other semantic text models can achieve comparable or even better results than traditional distributional semantic models. Models trained on domain-specific data are more effective than those trained on general language or unrelated subjects like Newtonian physics. Additionally, combining semantic text models with regular expressions enhances their performance, aligning them closely with human judges' evaluations. These insights have implications for improving the grading process in conversational ITSs and emphasize the need for further research in domain-specific areas.
Reviewer 2 Report
General comments:
The paper presents a comprehensive evaluation of various semantic text models in the context of grading student responses to electronics questions. The research addresses an important areas of interest, given the recent advanncements in text similarity models and their potential applications in educational technology. The study compares 13 different computational text models, including regular expressions, distributional semantics, embeddings, contextual embeddings, and combinations thereof, to assess their performance against expert human judges.
Strength:
The paper in general provides a concise overview of the study's objectives, methods, and key findings. Its abstract and introduction highlights the focus on domain-specific training corpora and the potential superiority of combination models involving both regular expressions and semantic text models in achieving high agreement with human graders. The paper effectively sets the stage for the research.
The Method and Results section describe with detail the key findings, emphasizing the success of the RegEx and Word2Vec combination model, which achieved results close to human-level agreement. The authors point out the importance of domain-specific training data for improving performance in the automatic scoring task. The comparison of large pre-trained transformer models with domain-specific models sheds light on the suitability of different models for specific tasks, and the preference for unsupervised methods with domain-specific corpora is well-supported.
Weakness:
There is one major drawback to me: The structure of this paper is really weird. The Introduction section is almost 9 pages long. Method and Results sections can be fairly short comparing to the introduction. I understand that this is in fact a review paper. However, such a long introduction is really strange. I hope the authors can re-organize the paper so that some details in the Introduction section can be transferred to Method or Results sections. I regard this as a major revision requirement.
Conclusion:
Overall, this paper makes a fair contribution to the assessment of semantic text models for automatic scoring in educational settings, particularly in the domain of electronics. The results are interesting: large pre-trained models sometimes cannot out-perform carefully designed domain-specific models. But the authors should indeed re-organize the structure of this paper in order to shorten the Introduction Section. I regard this as a major revision request.
The English presentation is fine. I did not find obvious grammatical errors.
Reviewer 3 Report
This paper focuses on the research about the ability of semantic text model to evaluate students' response to electronic problems. Based on RegEx and W2V model, the author analyzed its performance index.
However, the content given in this paper it has limited innovations. The studies in this paper lack of clear explanations and the practical significance needs to be further explained, better presented.
Concretely, there are several problems noticed in this paper, a list of these problems is given below. Please check the following list and make further discussion and revision seriously.
1 Problem: Key words should be further simplified to reflect the research characteristics of the article.
2 Problem: The current manuscript needs to be polished by a native English speaker or a professional language editing service.
3 Problem: The complexity and efficiency of the algorithm should be further analyzed to enhance the application value of this research.
4 Problem: At the INTRODUCTION, please be more specific about the motivation of this article. What are the shortcomings of the current relevant methods, or what are the advantages of the research content of this paper.
5 Problem: The current limitations as well as the respective characteristics of the related works are not described clearly. Please provide a table or a figure summarizing and comparing the main contributions and features of some other related researches and the key research variables of the approach presented in this paper.
6 Problem: The experimental design is too simple. The author did not compare the differences in some relevant studies, especially the latest research results, so as to highlight the effectiveness and innovations of the model proposed in this paper.
The current manuscript needs to be polished by a native English speaker or a professional language editing service.
Round 2
Reviewer 2 Report
Roughly speaking, the authors revised the paper to mitigate the weakness I pointed out last time. As a result, I recommend to accept this paper.
The English quality is fine. There is no need to further polish the paper for writing and grammars.